# Machine Learning Technique for Recognition of Flotation Froth Images in a Nonstable Flotation Process

## Jacek Galas * and Dariusz Litwin

Łukasiewicz Research Network, Tele and Radio Research Institute, 11 Ratuszowa Str., 03-450 Warsaw, Poland
* Correspondence: jacek.galas@itr.lukasiewicz.gov.pl; Tel.: +48-604-432-844

**Abstract:** The paper is focused on the analysis of the relation between the stability of the flotation process and the efficiency of Machine Learning (ML) algorithms based on the flotation froth images. An ML process should enable researchers to construct Artificial Intelligence (AI) algorithms for flotation process control. The image of the flotation froth includes information characterizing the flotation process. The information can be extracted with the aid of the Image Recognition (IR) algorithms based on the ML. This enables construction of a flotation process control system in the mineral processing plant, which is based on the recognition of images of the flotation froth. The IR algorithms do not provide stable image recognition results and are not efficient in the situation where the parameters of the flotation process are highly unstable. The classification results were equal to 75.11% and 69.62% for a stable and unstable process, respectively. The experimental data collected at the Polish Pb/Zn mineral processing plant provided better insight to the relationships between the flotation process parameters and ML efficiency. These relationships were analyzed, and guidelines for the construction of the ML process for flotation process control have been formulated.

**Keywords:** machine learning; image recognition; artificial intelligence; mineral processing; flotation technology





## 1. Introduction

The ore flotation process is of particular importance for mineral processing technology. This research aimed to evaluate mineral processing parameters and their optimizations. In the longer term, it should lead to better economic results of the process as a whole [1–3]. The technological and economic strategies for optimization of production plants have also been investigated [4].

The ore beneficiation process has been theoretically investigated in the case of the optimization of the flotation technology and the possibility of forecasting the results of the ore beneficiation process [5–9].

Another approach is directed toward the application of the ML process for extraction of the information about the status of the flotation process.

An image of the froth surface contains information about the characteristics of flotation processes. This information is rather complex and is related to the parameters of the technological process. The problem consists in extracting this information, especially in situations where the visual component is complex, and the useful content (signal) is mixed with information that can be treated as "noise". One problem to be solved is to find the relation between the froth's content and the froth's image parameters.

One of the first visual systems for flotation process control was implemented at the KGHM Polska Miedź S.A., Ore Enrichment Division [10,11]. The FloVis visual system captures the froth's images at the flotation machines, and following image processing, it calculates the image parameters including the size, shape, and color of froth bubbles, their mobility, and several other features. These are used subsequently to control the parameters of the flotation process. Apart from FloVis, it is possible to find other visual systems, for

example, the FrothSense+™ produced by Metso Outotec based on the older FrothSense™ and VisioFroth™ systems.

Even though the above approaches provided significant input to the flotation process control and were very important from an economic point of view, the image parameters were not linked to the froth's content. The knowledge of this relation could have a profound effect on the control of the flotation process and could improve the economy of the mineral processing plant. The technology to perform this task consists in advanced processing and deeper analysis of the froth image with the aid of statistical and ML algorithms.

In general, researchers prefer two main approaches to image processing software development to extract the information useful for the determination of the flotation process characteristics. In the first one, the information is extracted directly from the froth image [10–13] based on "visual" parameters such as the bubble size, shape, velocity, etc.

The second one is based on the Fourier transform of the froth image [14–23]. In this approach, the image parameters (descriptors) relate to the froth "visual" parameters (bubble size, shape, velocity, etc.), but these connections are not straightforward or simple. These parameters should be treated as statistical image descriptors and are very useful in statistical image classification/recognition processes.

The ML process provides the ability to find the relation between the froth's content and the froth's image parameters. It is based on the training groups of froth images, captured in various stages of the flotation process, where the froth's content varies according to the flotation parameters. The ML process leads to an Image Recognition (IR) algorithm, which enables establishing a relation between the froth's image and its content.

The images for the ML process should be first digitally processed and analyzed. The Image Analysis procedure must be sensitive enough to detect significant variation in the flotation process parameters [14–18]. The image recognition process has been described in [19,20]. A selection of applications of the IR algorithms on flotation technology was described in [20–23].

In general, the information characterizing the flotation process, coded in froth images, relates to the parameters describing the investigated industrial process. These parameters can be generated in various ways: constructed by software on the vision signal (image features) or collected by other sensors measuring, for example, air pressure in the flotation machine, dosage of chemicals such as the frother, etc. Therefore, the multisensor data fusion is very helpful [24] for all processes involving ML.

In our case, the ML process was built on the features extracted exclusively from the froth images, and the selection of these features is the most important factor for the effectiveness of the ML process. Exemplary feature selection methods are presented by Girish Chandrashekar et al. [25].

The ML process can be applied to build software tools to support the control of the ore technological beneficiation process at the mineral processing plant. It can also help us to understand some of the relations between the parameters of the flotation technology and the froth appearance in various stages of the technological process and the content of the flotation froth.

This paper presents the results obtained by the ML process, where the image features were constructed based on their Fourier transforms. In this way, the froth behavior was investigated in an unstable flotation process, where the froth content strongly varied in time. Although the ML process was not optimal in such circumstances, the image classification/recognition results were satisfactory. This indicates that the next step along this path is to find the algorithm for flotation process control in stabilized process conditions.

As stated in [23], the calibration of the experimental data, the measuring accuracy, and the protection of the system from industrial pollution (i.e., dust and chemical pollutants) are separate problems in the image analysis techniques developed for industry. The optical equipment is susceptible to vibrations, distorted inhomogeneous illumination, etc. Therefore, the influence of these factors should be eliminated.

## 2. Materials and Methods

### 2.1. Experimental Setup

The experimental flotation process was carried out in the sphalerite/galena processing plant [23]. The scheme of the flotation machine with the installed measurement system is presented in Figure 1.

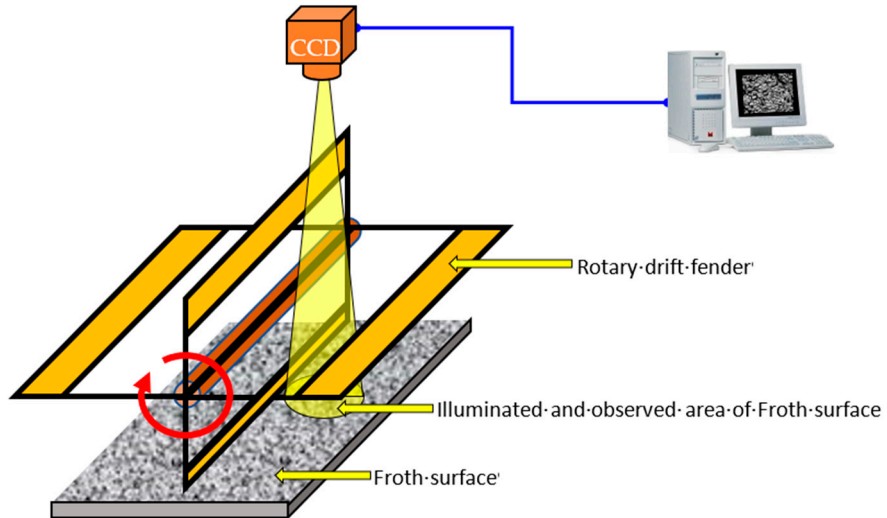

**Figure 1.** Scheme of the flotation machine with the installed measurement system.

The surface of the froth in the flotation machine was observed by the CCD camera ($512 \times 512$ pixels, 8 bits/pixel) mounted 30 cm above the froth surface at the cleaning flotation machine. The froth images from the illuminated and observed area, captured by the camera, were sent to the computer for further processing and analysis. The flotation process was controlled by the following parameters: the amount of collector, depressor, frother, promoter, feed, and drift fender revolutions. The image registration process was synchronized with the drift fender revolutions to avoid vignetting of the observation area.

The elimination of the vibrations of the CCD camera and lighting system was obtained by mechanical isolation of the measurement system from the flotation machine.

### 2.2. Image Registration Process

The images were registered in 8 main technological groups for various contents of froth. For each group, the technological parameters of the flotation process were stabilized. We expected that the Pb content variation within a group would not be high and the standard deviation of %Pb would not exceed 0.5 percentage point.

For each of 8 groups, 205 images were registered in 10 subgroups (the first nine subgroups contained 20 images and 10th subgroup contained 25 images), which means that, in total, 80 subgroups of images were registered. During the image registration in a subgroup, the flotation froth was continuously gathered to a container for chemical analysis. Each of the subgroups had a chemically determined %Pb, which was the mean value of the Pb content during the registration time of the images in a given subgroup.

The eighth group was not taken into consideration because the variation in time of the Pb content was too high for efficient image classification (its maximal variation—Max $\Delta$%Pb = 11.83%) during further analysis.

The flotation process was tuned to the requirements of the experiment. The aim of the tuning process was the stabilization of the flotation parameters in some period of time. This should have led to stabilization of the Pb content in the froth at the expected level during the image registration time. The parameters of the flotation process were published in [23]. In these conditions, we planned to register the images of the froth's surface. Figure 2 presents the plot of the Pb percentage content (%Pb) in the froth versus time during the

flotation experiment. The start and stop image registration times for the given group are marked by the color vertical lines. Each color dot represents the froth sample with chemically measured Pb percentage content. The sample was represented by 20/25 froth images captured by the camera. For each group, the following information is presented: groups registration times and the maximal difference in Pb content within the groups. The time periods between the registration of subsequent groups are also presented.

Figure 3 presents the plot of variation of %Pb versus time in the groups.

Contrary to what we expected, the flotation process was not stable. The variations in %Pb content in the froth within many groups were high and for 5 groups exceeded the assumed value. The variation in %Pb content in time was defined as $\Delta\%Pb/\Delta t$ where $\Delta t = 5$ min.

Table 1 presents the variations in Pb content for the first 7 groups. For a given group, the subsequent columns present the mean value and standard deviation of %Pb value as well as the mean value and standard deviation of $\Delta\%Pb/\Delta t$.

**Table 1.** Variations in %Pb content in groups.

| Gr. No. | Mean Value of %Pb | Std. Dev. Value of %Pb | Mean Value of $\Delta\%Pb/\Delta t$ | Std. Dev. Value of $\Delta\%Pb/\Delta t$ |
|---|---|---|---|---|
| 1 | 76.90 | 0.80 | 0.17 | 0.86 |
| 2 | 75.77 | 0.26 | −0.09 | 0.29 |
| 3 | 75.43 | 3.72 | 1.73 | 4.80 |
| 4 | 73.00 | 1.58 | 0.73 | 1.33 |
| 5 | 75.05 | 1.45 | 0.29 | 4.52 |
| 6 | 64.99 | 0.44 | −0.34 | 1.19 |
| 7 | 64.74 | 0.57 | −0.07 | 1.56 |

Considering the nature of the process, it was difficult to stabilize the Pb content in the froth. The flotation process has an "autoregulation" mechanism that tries to set the "optimal" froth content for the given parameters of the flotation process in flotation machines. We expected that the period between registrations of the groups would be long enough for froth content stabilization. However, this was not true for Groups 3, 4, and 5, as shown in Figures 2 and 3.

For the image groups 1–7, the ML process was performed, and its effectiveness was tested.

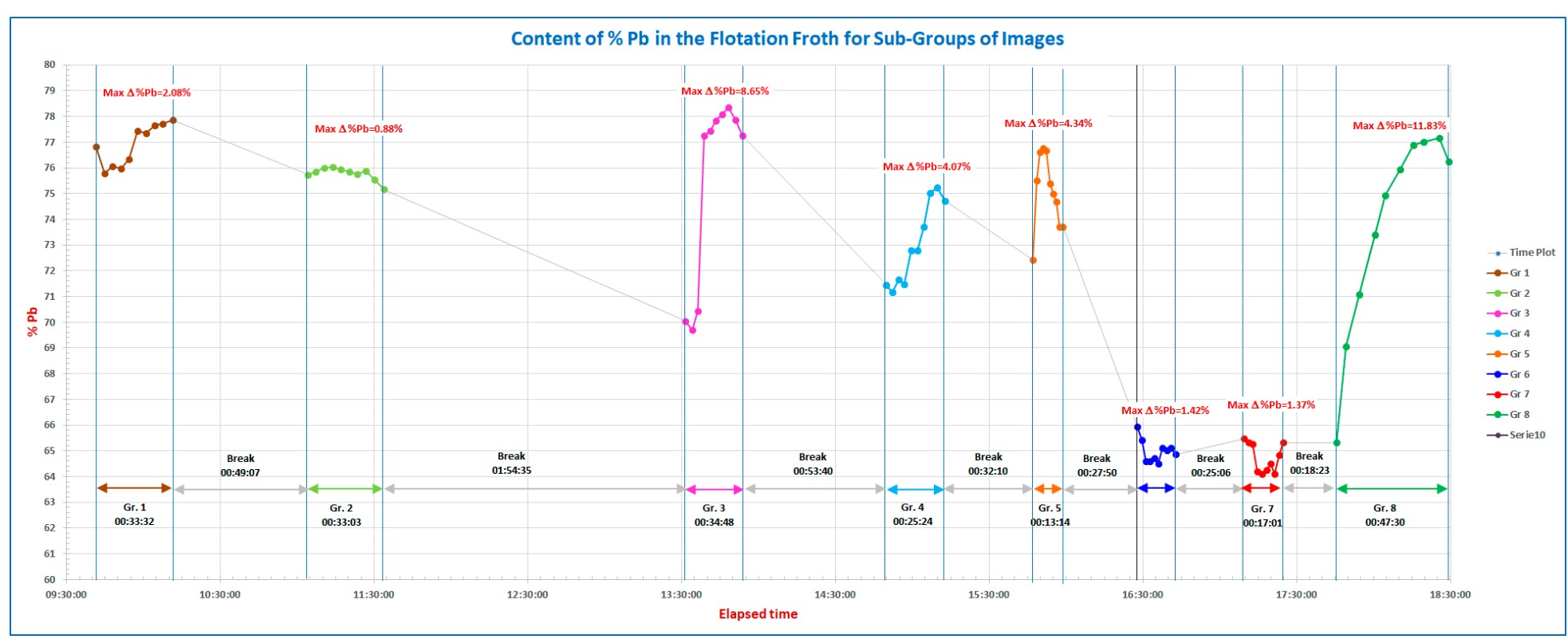

**Figure 2.** Content of %Pb in the flotation froth versus time during the experiment.

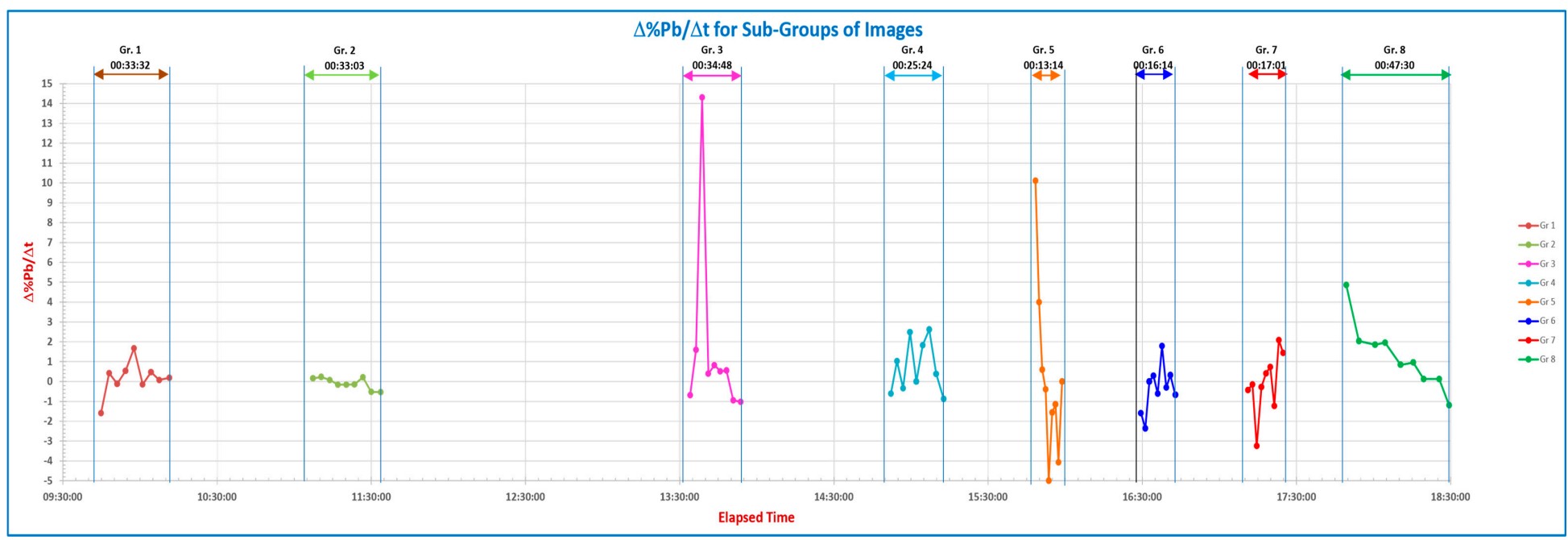

**Figure 3.** Variation in %Pb content versus time in the groups of flotation froth (Δ%Pb/Δt) during the experiment.

### 3. Results

*3.1. ML for Unstable Flotation Process*

The ML process can be performed in various ways. Several ML strategies were analyzed by Aldrich et al. [26], which provided insight into the issue. There are two main approaches to ML process implementation. The first can be performed with the aid of neural networks [27,28], and the second utilizes Linear Discriminant Analysis (LDA) [16,19–23]. This paper focused on the latter.

The Machine Learning process was performed for groups 1 to 7 that contained the training images. Based on the LDA, these images were used during the ML process to calculate the discriminant functions. Then, the classification algorithm was constructed. This process as well as the algorithm and criteria for image classification are presented in detail in [23]. We wanted to estimate the depth of the relation between the froth image and its content to determine whether it was possible to build an AI algorithm, based on the ML process, to estimate the froth content in the industrial conditions.

The LDA, as the part of ML process, was applied to construct the algorithm to classify each image into a specific group. The images to be classified were firstly transformed to their Fourier transform representation; then, a set of image parameters was constructed [23]. The set of parameters in the presented research included 96 parameters.

Table 2 presents the classification results for all images of the seven groups recorded during the experiment. The first column presents the number of a group defined in the experiment. The second column contains the number of images in each group. The next columns subsequently contain the percentage of images classified from a given group (1st column) to the other groups, which demonstrates to what extent the ML process was effective.

**Table 2.** Classification results for all images collected during the experiments.

| Classification Results for All Images | | | | | | | | |
|---|---|---|---|---|---|---|---|---|
| **Defined Group** | **No. of Images** | **Classification Groups** | | | | | | |
| | | **Gr_1** | **Gr_2** | **Gr_3** | **Gr_4** | **Gr_5** | **Gr_6** | **Gr_7** |
| **Gr_1** | **205** | **60.00%** | 24.88% | 4.88% | 10.24% | 0.00% | 0.00% | 0.00% |
| **Gr_2** | **205** | 20.49% | **63.41%** | 5.37% | 10.73% | 0.00% | 0.00% | 0.00% |
| **Gr_3** | **205** | 7.32% | 6.34% | **70.24%** | 9.27% | 3.90% | 2.93% | 0.00% |
| **Gr_4** | **205** | 7.32% | 5.37% | 13.17% | **74.15%** | 0.00% | 0.00% | 0.00% |
| **Gr_5** | **205** | 0.00% | 0.00% | 0.98% | 1.46% | **73.17%** | 20.00% | 4.39% |
| **Gr_6** | **205** | 0.00% | 0.00% | 1.46% | 0.00% | 7.80% | **73.66%** | 17.07% |
| **Gr_7** | **205** | 0.00% | 0.00% | 0.49% | 0.00% | 1.46% | 25.37% | **72.68%** |
| **Number of images correctly classified: 69.62%** | | | | | | | | |

Interestingly, the classification results strongly depended on the process parameters, not only on the %Pb content in the froth, as was expected. The speed of %Pb changes during the time of group registration was particularly important. The total number of images that were correctly classified was equal to 69.62%.

In the case of relatively similar %Pb, in some groups, a significant number of images was misclassified i.e., 40% and 36.59% in Groups 1 and 2, respectively. In order to avoid this problem, it would be necessary to extend the set of image parameters, which should be the subject of future studies.

*3.2. ML for Stable Flotation Process*

To increase the number of correctly classified images, some of them were excluded from the ML process. The plot of Pb content versus time is presented in Figure 4. The excluded subgroups are marked by red ellipses.

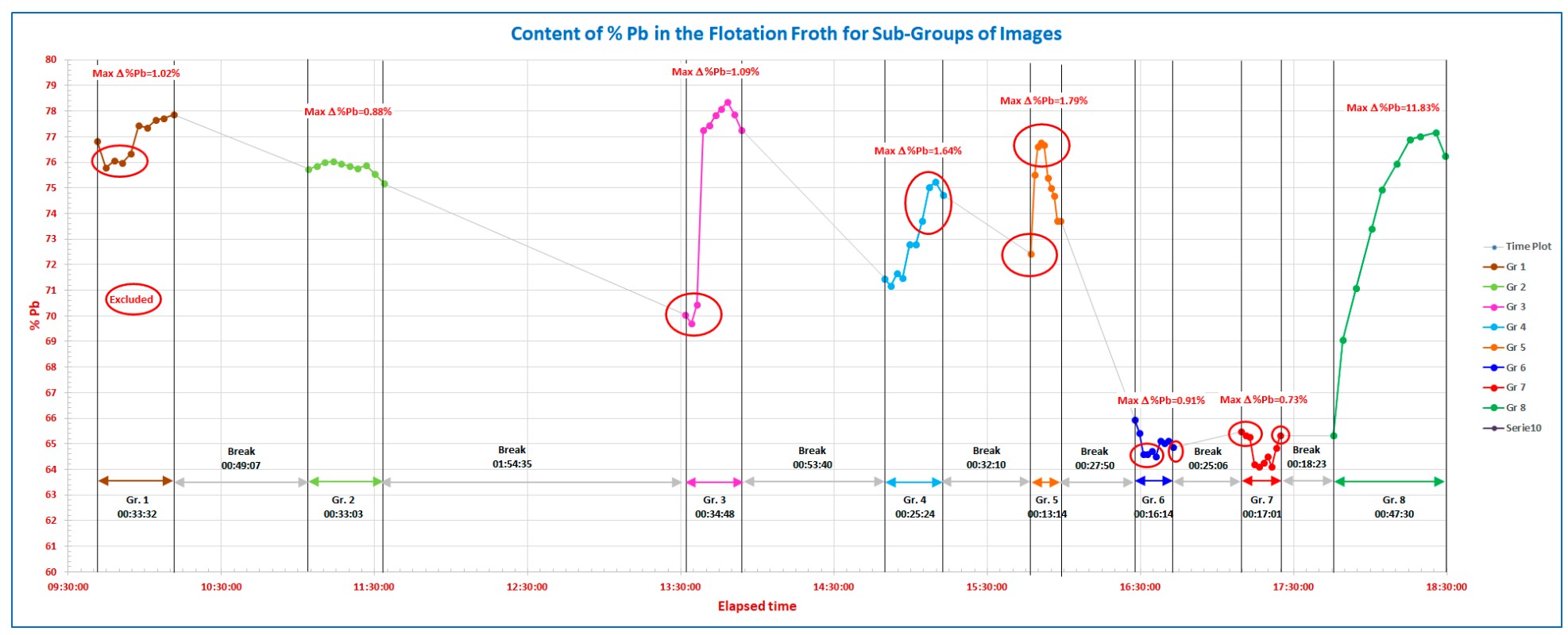

**Figure 4.** Content of %Pb versus time in the flotation froth.

As mentioned above, the froth contents within each group were not stable, and the %Pb in many groups varied significantly. The aim of the data exclusion process was to suppress the range of %Pb variations within the group and to differentiate the froth content among adjacent groups.

For Group 1, there were excluded points for which the froth content was very close to the content in Group 2. The %Pb variation was reduced from 2.08% to 1.02%.

Group 2 had a %Pb variation below one percentage point, and we left this group without any changes.

For Group 3, we excluded three points of the highest difference in the %Pb and retained the rest of points. The %Pb variation was reduced from 8.65% to 1.09%.

For Groups 4 and 5, we excluded the points for the maximal differentiation of the froth content between these groups. It also allowed us to reduce the %Pb variation from 4.07% to 1.64% (Group 4) and from 4.34% to 1.79% (Group 5).

A similar situation occurred for Groups 6 and 7. The exclusion of the points enabled us to differentiate the froth content between these groups and to reduce the %Pb variation from 1.42% to 0.91% (Group 6) and from 1.37% to 0.73% (Group 7).

For the modified groups, the variation in %Pb content within groups is presented in Table 3. The elimination of the marked subgroups decreased the %Pb dispersion within groups below 0.8 percentage point.

**Table 3.** Variations in %Pb content in the modified groups.

| Gr. No. | Mean Value of %Pb | St. Dev. Value of %Pb |
|---|---|---|
| 1 | 77.48 | 0.36 |
| 2 | 75.77 | 0.26 |
| 3 | 77.73 | 0.42 |
| 4 | 71.89 | 0.72 |
| 5 | 74.67 | 0.79 |
| 6 | 65.32 | 0.17 |
| 7 | 64.33 | 0.28 |

Table 4 presents the classification results for the images after the exclusion of the subgroups. After the exclusion process, the number of correctly classified images significantly rose to 75.11%.

**Table 4.** Classification results for the modified groups (subgroups excluded).

| Classification Results for Images after Exclusion | | | | | | | | |
|---|---|---|---|---|---|---|---|---|
| Defined Group | No. of Images | Classification Groups | | | | | | |
| | | Gr_1 | Gr_2 | Gr_3 | Gr_4 | Gr_5 | Gr_6 | Gr_7 |
| Gr_1 | 125 | 52.80% | 37.60% | 3.20% | 6.40% | 0.00% | 0.00% | 0.00% |
| Gr_2 | 205 | 14.63% | 78.05% | 0.98% | 6.34% | 0.00% | 0.00% | 0.00% |
| Gr_3 | 145 | 4.83% | 3.45% | 83.45% | 4.14% | 2.76% | 1.38% | 0.00% |
| Gr_4 | 120 | 6.67% | 10.00% | 5.00% | 78.33% | 0.00% | 0.00% | 0.00% |
| Gr_5 | 125 | 0.00% | 0.00% | 4.80% | 0.00% | 73.60% | 19.20% | 2.40% |
| Gr_6 | 100 | 0.00% | 0.00% | 3.00% | 0.00% | 8.00% | 74.00% | 15.00% |
| Gr_7 | 120 | 0.00% | 0.00% | 0.00% | 0.00% | 0.83% | 16.67% | 82.50% |
| Number of images correctly classified:75.11 | | | | | | | | |

However, as observed earlier, the higher number of misclassified images belonged to the modified Group 1. This confirmed that the image parameters should be reconsidered, to distinguish Group 1 from 2, where the flotation process parameters differed from each other. Interestingly, the modified Groups 1 and 3 had a similar mean value of %Pb, but

they were distinguished, and the numbers of misclassified images between these groups were very low. This also is planned to be a subject of further research.

## 4. Conclusions

The results of the described experiment demonstrate that the prediction of the froth content based on the froth surface images using an ML algorithm is possible and successful. This algorithm was based on the described image processing and analysis techniques. The ML algorithm can lead to the construction of an efficient AI algorithm that will connect the information about the froth image and flotation process parameters with the froth content.

The ML process was more effective in the case of a stable flotation process where the variation in the Pb content was lower. Stabilization of the froth content at the level of 1 percentage point of %Pb during the registration of the training group of images significantly enhanced the effectiveness of the image classification algorithm and consequently the effectiveness of the forecasting process.

The variation in Pb content in time ($\Delta\%Pb/\Delta t$) influenced the effectiveness of the ML process in various ways. This should be considered in connection with the flotation process parameters. For some combinations of these flotation process parameters, even relatively high values of $\Delta\%Pb/\Delta t$ (st. dev. of $\Delta\%Pb/\Delta t$ was equal to 1.19 and to 1.56 for Group 6 and 7, respectively) did not reduce the effectiveness of the classification algorithm (see Tables 1–3). On the other hand, the relatively low value of this parameters (std. dev equals to 0.26 for Group 2) did not improve the classification results. The issue should be central for further research.

However, even in the conditions of the presented experiment, the information about the froth content was extracted from the registered images of the froth surface at a satisfactory level.

The relation between the froth image parameters and flotation process parameters should be the topic of further investigation; however, the experimental results showed that a relation can be already seen in most of the observed cases, which is promising.

**Author Contributions:** Conceptualization, J.G.; methodology, J.G. and D.L.; data processing, J.G. and D.L.; writing—original draft preparation, J.G.; validation, D.L.; writing—review and editing, D.L. All authors have read and agreed to the published version of the manuscript.

**Funding:** The research was financed by the institute statutory funds, granted by the Łukasiewicz Center, No. 27/CŁ-Inst./2022.

**Acknowledgments:** The article is the result of the statutory project entitled: „Techniki przetwarzania, rozpoznawania i analizy obrazów w zastosowaniach przemysłowych z wykorzystaniem sztucznej inteligencji" ("Industrial applications of image processing, recognition, and analysis with the aid of an artificial intelligence technique"). The authors would like to thank Stanisław Lenczowski and Jacek Kordek, AGH University of Science and Technology, Cracow, Poland for their earlier research contribution during cooperation with the Maksymilian Pluta Institute of Applied Optics, Warsaw, Poland, in the field of image recognition of flotation froth.

**Conflicts of Interest:** The authors declare no conflict of interest.

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
