# Peer review of "Machine Learning Technique for Recognition of Flotation Froth Images in a Nonstable Flotation Process"

_minerals, doi:10.3390/min12081052_

Round 1
Reviewer 1 Report
This is a valuable survey on the application of ML technique to metallurgically assess a Pb/Zn flotation process. The concept behind this effort is interesting however, there some key technical and scientific flaws challenging the quality of this work that esteemed authors should consider in their modified paper:
1. Please furnish the abstract with important numerical results.
2. Methodology is obscure. Please use suitable subsections to give more details on the used methodology, algorithms applied, calibration technique, hardware configurations etc.
3. Discussions in this for is not acceptable. Authors may be suggested to paint their discussions by giving some comparative words in comparison with other image analysis works.
4. Conclusions should be edited.
5. Please improve the quality of images.
Reviewer 2 Report
Dear authors! I read your article with pleasure. I hope that my review will help make it even more interesting for readers.
The development of operational control methods of ore flotation is an important scientific and practical task. Modern flotation factories use video image analysis of flotation foam. The vision system is able to control the characteristics of the foam layer, including the rate of foam descent, bubble size, foam stability and color. Image analysis must adapt to the changing process, so it is necessary to collect, accumulate and analyze information constantly. The purpose of the article is to analyze the effectiveness of machine learning algorithms depending on the stability of the flotation process. It seems reasonable and important. The strong point of the article is the presentation of experimental data collected at the plant, and a detailed description of their analysis. The weak side of the article is the lack of a description of the machine learning method that should be presented in the article in accordance with its title (perhaps the title should be changed); the lack of conclusions and recommendations on building an ML process to control the flotation process. The triviality of the conclusions drawn. The absence of specific parameter values in the description of the results obtained and the conclusion.
The issue discussed in the article is important. The analysis made by the authors is indicative. It shows the problems of improving the accuracy of flotation foam image analysis, which makes it possible to identify new tasks for future research.
The main text of the article (excluding abstract and list of articles) is less than 4000 words and can be added. The topic of the article corresponds to the section Mineral Processing and Extractive Metallurgy.
The results presented in the article are interpreted correctly. The conclusions are consistent with the results. But in the description of the results there is no necessary concretization and detailing. The authors use the expressions "was too high" (line 120), will not be high (line 131), "were high variation" (line 142). It is desirable to explain the high performance is higher than what value? Give specific values for the upper bound or interval. The authors of the article did not formulate a specific hypothesis. Initially, it was planned to study under conditions of stabilization of the quality of the foam. But from the text of the article, it can be assumed that the hypothesis could be that at a certain high level of variation in the content of lead in the flotation foam (% Pb / t), the efficiency of the forecast on the image of the foam decreases.
However, the text could be improved. The authors' decisions are not always clear and obvious from the text. For example, it is not clear why these particular points were excluded from the analysis (Fig. 4). What was the exclusion criterion?
If the 8th group was not analyzed because the change in foam content over time was too large for efficient image classification, then there are limitations that can be indicated.
It would be interesting to see images of foam with minimum and maximum lead content.
In general, the results are presented quite fully and clearly. The interpretation of the results is clear. Graphs and tables are clear. But figures 2,3,4 are poorly perceived because of their location.
Link to source 23 requires verification.
1. There is no similar figure to Figure 1 in the source
2. The paragraph (lines 101-105) has no reference to source 23, but the text of the paragraph is taken from this source.
3. In source 23 there is no mention of 96 parameters, which are written in line 176.
The study is constructed in accordance with the accepted methodology for analyzing the reliability of the information received, studying the relationship between the foam image and flotation technological indicators. The linear discriminant analysis widely used for the interpretation of video images is used, which ensures the reliability of the results.
The shooting conditions are described by the authors in general terms. The height of the camera placement above the surface of the foam layer, the speed of rotation of the foam pad, how the effect of vibration and inhomogeneous lighting was reduced during the experiment are not specified.
There is no description and image of the foam layer itself.
There is no flotation scheme indicating the location of the camera, the flotation operation is not specified (main flotation, perechist flotation ...). All this reduces the probability of reproducing the experiment by other researchers.
The algorithm and criteria for image classification are not presented.
The article will be of interest to specialists because it complements statistical information about the influence of fluctuations in the content of a valuable component in foam on the results of forecasting based on images of the foam surface using artificial intelligence algorithms.
When reading the article, a question arose. The condition of the foam layer, the mineralization of the foam during the stabilization of the parameters of the flotation process controlled by you depends on the quality of the feedstock. Have you carried out quality control of the source ore?

Round 2
Reviewer 1 Report
The paper is now suitable for publication.
Reviewer 2 Report
Dear authors! After the clarifications and additions you have made to the article, the conclusions look more reasonable. It is obvious to me that you will be able to answer some of my comments only after conducting research in the future. Your article shows the existing problems of creating machine learning algorithms on real data in conditions of low process stability. The experience you described and the results you obtained are useful for future research in this direction. I wish you a successful continuation of your researc.